# Effect of Shade Screen on Sap Flow, Chlorophyll Fluorescence, NDVI, Plant Growth and Fruit Characteristics of Cultivated Paprika in Greenhouse

**Kyeong Ho Kim** [1,†]**, Md Rayhan Ahmed Shawon** [2,†]**, Jin Hee An** [1]**, Hyoun Jin Lee** [1]**, Dong Jae Kwon** [3]**, In-Chul Hwang** [4]**, Jong Hyang Bae** [5] **and Ki Young Choi** [1,2,*]

1  Department of Agriculture and Industries, Kangwon National University Graduate School, Chuncheon 24341, Korea
2  Division of Future Agriculture Convergence, Department of Controlled Agriculture, Kangwon National University, Chuncheon 24341, Korea
3  Catalonix, Director R&D Lab, A-906,114 Beopwonro, Songpa-gu, Seoul 05854, Korea
4  Department of Electrical and Electronics Engineering, Kangwon National University, Chuncheon 24341, Korea
5  Department of Horticulture Industry, Wonkang University, Iksan 54538, Korea
*  Correspondence: choiky@kangwon.ac.kr; Tel.: +82-10-8984-9646
†  These authors contributed equally to this work.

**Abstract:** The aim of this study was to investigate the effect of shade screens on the physiological activity, growth parameters and fruit characteristics of the paprika (*Capsicum annuum* L.) plant. Plants were grown in a protected greenhouse and treated under two different shade screens, S1 (single screen) and S2 (double screens; 10% low light intensity compared to S1), during summer at a particular time of the day. The results revealed that the plant height was significantly enlarged by the S2 treatment. However, the number of leaves, leaf fresh weight and leaf dry weight were significantly decreased under S2-treated plants compared to those grown in the S1 treatment. The stem diameter and shoot fresh weight were not significantly different between the treatments. The sap flow and normalized difference vegetation index (NDVI) were higher in S1-treated plants than in those grown in the S2 treatment. The chlorophyll fluorescence fluctuated in both treatments. The fruit fresh weight, number of fruits, fruit pericarp thickness, fruit firmness, fruit volume, sugar content and acidity were significantly higher in S1-treated plants than in S2. Hunter values *a* and *b* were significantly higher in S2-treated plants. Moreover, the fruit length and width were not significantly different between the two treatments. The sugar content and acidity of paprika showed a positive correlation. These results suggest that, compared to a double screen for shade in the greenhouse, a single screen is suitable for the growth of paprika plants and enhanced their fruit production.

**Keywords:** leaf temperature; light intensity; number of leaves; number of fruits; fruit firmness; Hunter value

## 1. Introduction

Paprika (*Capsicum annuum* L.) is a mostly consumed vegetable and used as a food colorant [1,2]. It is a rich source of carotenoid pigments and other phytochemicals, such as ascorbic acid, phenolic compounds and flavonoids [3], which prevent chronic diseases such as cardiovascular disease [4]. In particular, capsanthin and capsorubin are unique compounds in red paprika and have shown anti-oxidative and antitumor activities [5,6]. Today, paprika is one of the most economically important vegetables, and its demand is increasing day by day. All over the world, around 34.5 million tons of fresh paprika were produced in 2016, which was 25% greater than in 2006 [7].

Paprika has been one of Korea's leading export vegetables in the horticulture industry for the last decade [8]. In the early 1990s paprika was introduced in Korea [9]. Notably, paprika production in Korea rose sharply from 2000 to 2017 and was from 7500 tons to

78,000 tons. Furthermore, the Japanese paprika market now greatly depends on Korean paprika. Korea supplies around 78% of the total imported paprika by Japan in 2018 [10]. Nowadays, in Korea, it attracts many agricultural investors, and the cultivated area is increasing because of its export-oriented production. The greenhouse is considered useful for a better crop growth of paprika in protected facilities in Korea's winter and spring seasons. It provides the opportunity for growers to maintain an optimum environment and reduces the deviations in plant growth, fruit quantity and quality [11].

Healthy and vigorous plants are required by growers for profitable paprika production. In addition, the color and fruit shape are important indicators of the mature fruit quality of paprika for consumers and also regulates the market value [12]. During the summer season (from June to August), plants face high light irradiance stress in both open fields and greenhouses in Korea. As a result, the temperature increases and the relative humidity also changes inside the greenhouse. The speed of photosynthesis in paprika plants in a greenhouse was reported to decrease due to high-temperature stress [13]. In Korea, paprika growers generally use two layers of shade screens in the greenhouse to protect plants from scorching sunlight in summer. The installation of shade screens inside of the greenhouse is a little costly. Furthermore, shade screens influence the paprika plant's growing environment (light, temperature, humidity, etc.).

Shade screens mainly protect plants against sunlight; however, the physiology and structure of plants are greatly influenced by light [14,15]. In addition, light conditions greatly influence the color and shape of a fruit [16,17]. In particular, the photosynthetic activity of a plant completely depends on its received light intensity [18]. Researchers showed that a low light intensity inhibits plant productivity by affecting gas exchange [19], whereas a high light intensity has detrimental impacts on the photosynthetic apparatus [20]. Furthermore, leaves are a specialized organ of a plant for photosynthetic activity, and their development is complex and is affected by light [21].

Nowadays, sap flow, chlorophyll fluorescence and the normalized difference vegetation index (NDVI) are helpful in understanding the physiological activities of a plant. Sap flow in a plant can be used as an indicator of its water status [22]. Chlorophyll fluorescence is an effective tool for the measurement of photosynthetic metabolism. It is a quick and non-destructive method [23]. NDVI is one of the widely used technologies in the field of remote sensing and has a strong relationship with morpho-physiological variables, such as leaf health, leaf area index (LAI), biomass, plant productivity and chlorophyll concentration. NDVI helps to identify the health condition of plants based on plant reflections against the light at certain frequencies (some waves are absorbed and some are reflected) [24,25]. Although NDVI imagery is mainly utilized by space-borne (satellites) and air-borne (drones) systems, there are many advantages to using ground-based NDVI in terms of temporal and spatial resolution. Hence, in this study, in order to exploit the advantages of ground-based NDVI, time series NDVIs were obtained using the digital camera according to the top and bottom sections of the plants.

Suitable and cost-effective shade screen installation for paprika cultivation in a greenhouse is required by the paprika growers. Regarding paprika grown in the greenhouse, a few studies are available [26–29]. However, limited research has found an association with the impact of shade screens mainly on plants' vegetative to reproductive stage [30,31]. Therefore, this study was conducted in a greenhouse to investigate paprika plant development, in association with sap flow activity, chlorophyll fluorescence, NDVI, fruit characteristics and yield, as affected by single-layer and double-layer shade screens during the growing season.

## 2. Materials and Methods

### 2.1. Plant Material and Growing Conditions

A springtime-growing cultivar of red paprika (cv. Nagano, Rijk Zwaan Co., De Lier, The Netherlands) was cultivated in a greenhouse (4290 m$^2$) located in Inje-si, Gangwon-do, Korea (latitude, 38°06′ N; longitude, 128°17′ E). Seeds were sown on 4 July 2021 in a

plug tray filled with commercial growing media, and then seedlings were transplanted on 28 February 2021 into coir slabs (Coire badge, Happy Farmers, 120 cm × 15 cm × 8 cm) for growing.

### 2.2. Treatment

Plants were treated by two treatments, S1 and S2, from 1 June 2021 to 15 August 2021, which were implemented by a number of shade screens during the day from 11 a.m. to 3 p.m. inside the greenhouse (at 6 m above ground level). The rest of the time, these shade screens were not activated in the greenhouse. In S1 treatment, a single shade screen (LS Harmony 5215O, Svensson, Seongnam-si, Gyeongi-do, Korea) was activated when outside light intensity was 700 Wm$^{-2}$ or higher. On the other hand, double shade screens were activated in the S2 treatment by the combination of light and temperature. When outside light intensity was 700 Wm$^{-2}$ or higher upper screen (XLS 10 ultra-firebreak, Svensson, Seongnam-si, Gyeongi-do, Korea) was activated, and, additionally, when the temperature was also more than 28 °C, then lower screen (XLS 18 firebreak, Svensson, Yongin, Korea) was activated. The treated greenhouse area was 36 m × 52 m for both S1 and S2. The amount of light outside the greenhouse was measured on a 5 min basis using the environmental control program (MAXIMIZER 4.2.0. build 4771 Version, Priva B.V., De Lier, The Netherlands). Light, temperature and humidity (RH%) in both treatments were measured using independent data loggers (Watchdog 100, Spectrum Technologies, Inc., Plainfield, IL, USA) (Figure 1).

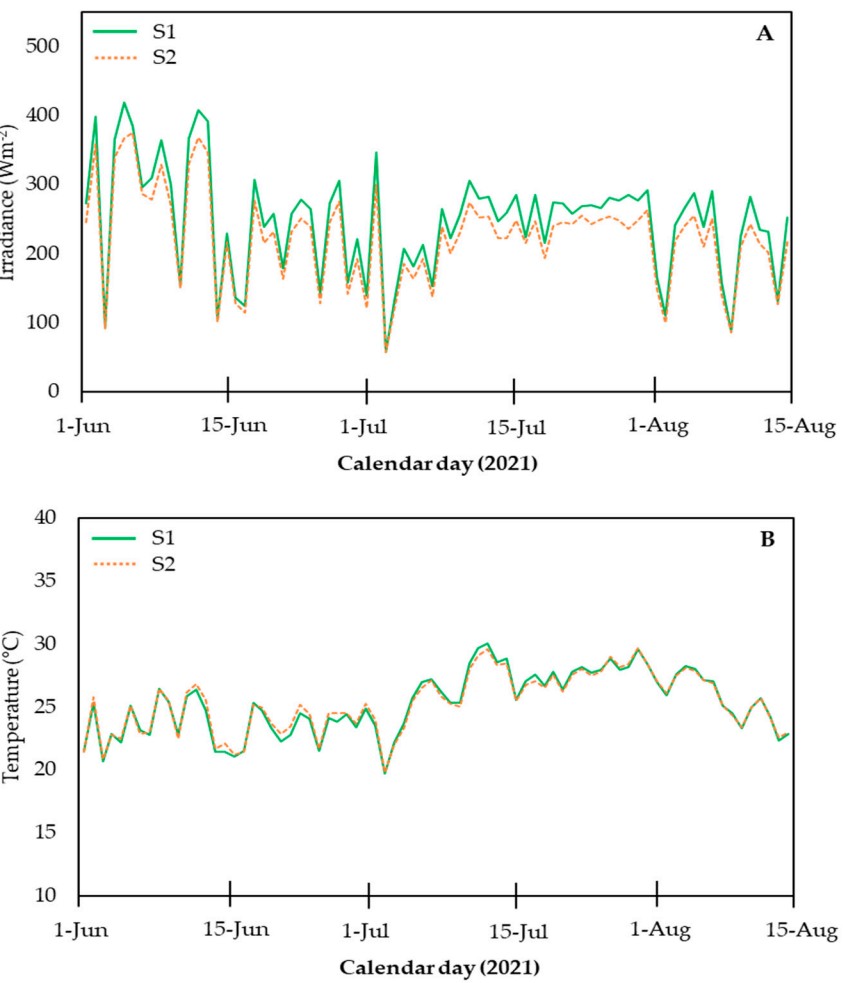

**Figure 1.** *Cont.*

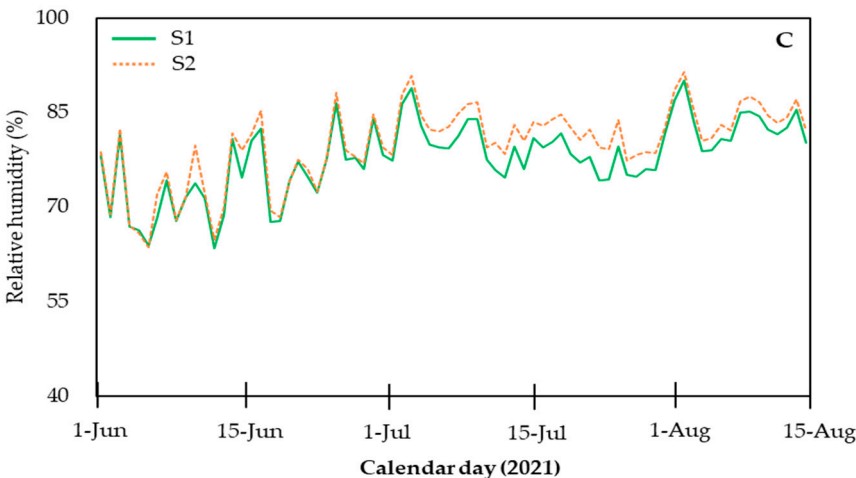

**Figure 1.** The average solar irradiance (11 a.m. to 3 p.m.) (**A**), average daily temperature (**B**) and relative humidity (%) (**C**) in both S1 and S2 sections of the greenhouse during the treatment period. S1 and S2 indicate single shade screen and double shade screens respectively.

The temperature of floors, plants and roofs in the greenhouse was measured using an infrared thermal imaging camera (FLIR-E63900, FLIR System Co., Stockholm, Sweden) at 9 a.m., 1 p.m. and 5 p.m. Leaf temperature of the plant was measured using a leaf temperature sensor (LT-1M, Bio instruments S.R.L., Chisinau, Moldova) at 1 m and 2 m height of the plant from the growing media. Moisture content in the substrate was measured by using a substrate weight sensor (IReIS, RMFarm, Gangneung-si, Gangwon-do, Korea) at 1 min intervals on 21 July 2021.

*2.3. Sap Flow Measurement*

Sap flow of the plant was measured by installing an SF-5M Sap flow sensor (Bio instruments S.R.L., Chisinau, Moldova) on 21 July 2021 in the stem (6–7 mm diameter), which is two or three nodes below the growing point, and data were collected in 1 min intervals.

*2.4. Chlorophyll Fluorescence Measurements*

Fluorescence measurements were recorded from fully expanded leaves of the plant's bottom portion (between node number 10 and 14 from the growing point) and top portion (between node number 18 and 22 from the growing point) using a portable fluorimeter (PAR-FluorPen FP 110/D, Photon Systems Instruments, Drásov, Czech Republic) at three-hour intervals from 8 a.m. to 5 p.m. on 21 July 2021. For the fluorescence readings, the leaves were previously dark and adapted for 30 min using the fluorometer leaf clips. Following dark adaptation, chlorophyll fluorescence was examined by the internal LED blue light (470 nm), producing a saturating light pulse of 2400 μmol photons $m^{-2}s^{-1}$, and the fast rise of chlorophyll fluorescence was recorded using the fluorimeter OJIP protocol (Table 1). Absolute values of chlorophyll fluorescence intensity are given in arbitrary units (a.u.).

**Table 1.** Equations and definitions of OJIP parameters (modified from Stirbet and Govindjee [32]).

| Parameter | Mathematical Equation | Description |
|---|---|---|
| | *Fluorescence transient OJIP* | |
| $F_o$ | $F_{50\mu s}$ | First reliable fluorescence value after the onset of actinic illumination; used as initial value of the fluorescence |
| $F_j$ | $F_{2ms}$ | Fluorescence value at 2 ms (J-level) |
| $F_i$ | $F_{30ms}$ | Fluorescence value at 30 ms (I-level) |
| $F_m$ ($=F_p$) | | Fluorescence value at the peak of OJIP curve; maximum value under saturating illumination |
| | *Technical fluorescence parameters* | |
| $F_v$ | $F_m - F_o$ | Maximum variable Chl fluorescence |
| $V_j$ | $(F_j - F_o)/(F_m - F_o)$ | Relative variable fluorescence at the J-level |
| $V_i$ | $(F_i - F_o)/(F_m - F_o)$ | Relative variable fluorescence at the I-level |
| $F_m/F_o$ | | Representing quantum yield of PSII photochemistry |
| $F_v/F_o$ | $(F_m - F_o)/F_o$ | Maximum primary yield of photochemistry of PSII |
| $F_v/F_m$ | $(F_m - F_o)/F_m$ | Maximum quantum efficiency of PSII |
| $M_o$ | $(\Delta V/\Delta t)_o = 4\,ms^{-1} \times (F_{300\mu s} - F_o)/(F_m - F_o)$ | Slope at the beginning of the transient $F_o \to F_m$, maximal fractional rate of photochemistry |
| | *Quantum yields and efficiencies/probabilities* | |
| $\Phi_{Po}$ | $TR_o/ABS = 1 - (F_o/F_m)$ (or $F_v/F_m$) | Maximum quantum yield of primary PSII photochemistry |
| $\Psi_o$ | $ET_o/TR_o = 1 - V_j$ | Probability that a trapped exciton moves an electron into the electron transport chain beyond QA |
| | *Specific energy fluxes (per active PSII reaction center)* | |
| $ABS/RC$ | $(M_o/V_j) \times (1/\Phi_{Po})$ | Absorption flux per RC |
| $TR_o/RC$ | $M_o/V_j$ | Trapped energy flux per RC (at $t = 0$) |
| $ET_o/RC$ | $(M_o/V_j) \times \Psi_o$ | Electron transport flux from $Q_A$ to $Q_B$ per RC (at $t = 0$) |
| $DI_o/RC$ | $(ABS/RC) - (TR_o/RC)$ | Dissipated energy flux per RC (at $t = 0$) |
| | Performance index (combination of parameters) | |
| $PI_{ABS}$ | $(RC/ABS) \times [\Phi_{Po}/(1 - \Phi_{Po})] \times [\Psi_o/(1 - \Psi_o)]$ | Performance index (PI) on an absorption basis (= energy conservation from photons absorbed by PSII antenna to the reduction in $Q_B$) |

*2.5. Measurement of NDVI*

The cameras were installed at similar geometry of S1 and S2 treatments to minimize unexpected variables, and time series of raw images were collected from 9 a.m. to 5 p.m. at 30 min intervals from 5 August to 15 August 2021. The raw images were obtained by IR-cut filter disabled digital cameras (Raspberry Pi NoIR Camera V2, Sony IMX 219, Tokyo, Japan) with a dual-bandpass (transmits 660 nm, 850 nm) filter (CATALOSCOPE, Catalonix Inc., Seodaemun-gu, Seoul, Korea) to obtain red and NIR signals (Figure 2A). Due to the absence of the IR-cut filter and the presence of the dual-bandpass filter, the B and G components of the charge-coupled device (CCD) receive only the NIR (850 nm) signal, whereas the R component receives both NIR and red (660 nm) signal.

Since the digital numbers (DN) value of the R component ($DN_{Red'}$) receives both NIR and red incident energy, it is possible to extract the contribution of the red signal ($DN_{Red}$) by multiplying the empirically obtained coefficient (k) (Figure 2B). Note that the DN is related to electrical responses that vary with incident energy, such as exposure time and the intensity of light sources, and not the reflectance that is spectroscopically useful. Therefore, a reference plate that has approximately 100% reflectance made by polytetrafluoroethylene (PTFE) was installed within the angle of view of each camera to convert raw images to reflectance images. For example, dividing the raw DN array of the B channel ($DN_{Blue}$) by the mean DN value of the reference plate pixels in the B channel ($DN_{RP(Blue)}$) gives reflectance in the B channel ($Reflectance_{Blue}$), which is identical to $Reflectance_{NIR}$. Based on the aforementioned reflectance images, NDVI was calculated using the following equation:

$$NDVI = (R_{NIR} - R_{red})/(R_{NIR} + R_{red}) \tag{1}$$

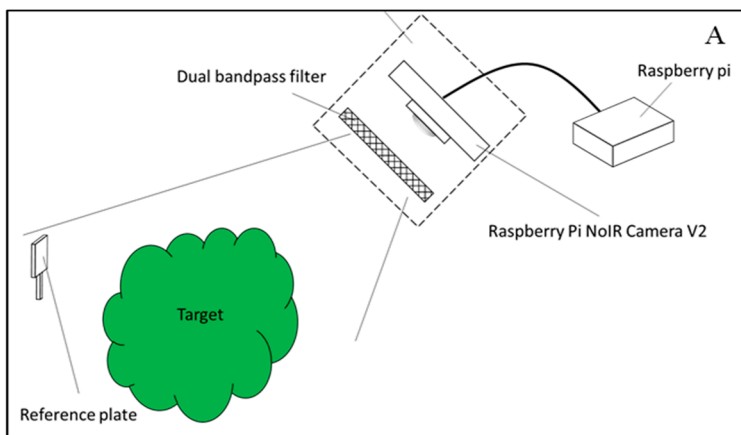

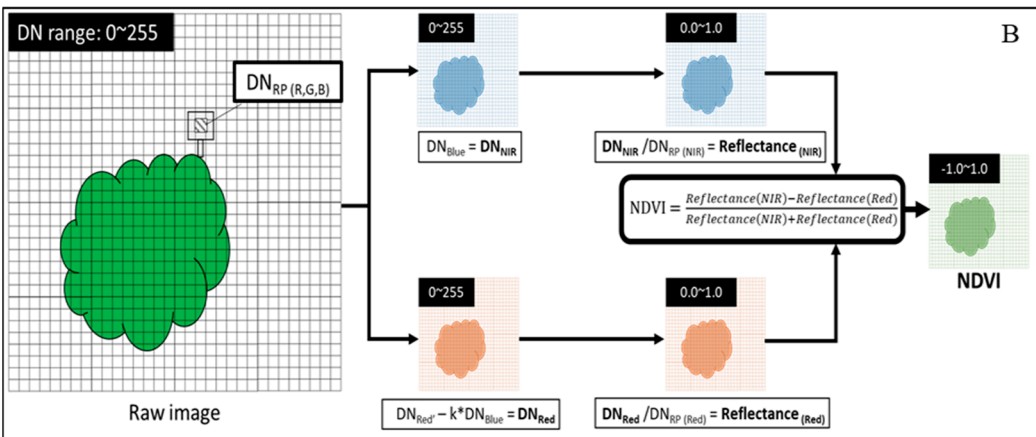

**Figure 2.** Camera setting (**A**) and calculation process (**B**) of NDVI.

Theoretically, NDVI values could be between −1 to 1, and a higher value means "healthy status" whereas a lower value indicates "unhealthy", dead or non-plant objects. Considering the relationship between NDVI and the potential phenological factors in this study, we could assume that the higher index indicates a higher growth rate (relatively higher LAI and chlorophyll concentration), and the lower index indicates a lower growth rate (relatively lower LAI and chlorophyll concentration).

### 2.6. Measurement of Plant Growth Parameters

Plants were uprooted on 31 October 2021 and the plant height, number of leaves, stem diameter, shoot fresh weight and leaf fresh and dry weight were measured. Plant height was measured using a tape ruler. The number of leaves was counted manually (less than 5 cm were excluded). A digital caliper (CD-20APX; Mitutoyo Corp., Kawasaki, Japan) was used for stem diameter measurement at 30 cm upper from the base of the plant. An electronic balance was used for leaf fresh weight and shoot fresh weight measurements. Leaf dry weights were gained after oven drying at 80 °C to a constant weight.

### 2.7. Measurement of Fruit Growth Parameters

Paprika fruit was harvested from 15 June 2021 to 30 August 2021. The numbers of fruit were counted manually (excluding deformed ones and less than 100 g) and paprika fresh weight was measured using an electronic balance. Fruit length and width were measured using a ruler. Fruit diameter was measured by a digital caliper (CD-20APX; Mitutoyo Corp., Kanagawa, Japan).

The external color of the paprika was measured (Hunter value) at room temperature by using a colorimeter (TES 135A, Shenzhen Youfu Tools Co., Ltd., Taiwan, China). The *L*

value indicates lightness; the *a* value indicates color red to green, where positive value (+) for red intensity; and the *b* value indicates color yellow to blue, where positive value (+) for yellow.

The firmness of paprika was measured at room temperature ($25 \pm 2$ °C) by a digital fruit firmness tester FR-5105 (Lutron electronic enterprise Co. Ltd., Taipei, Taiwan) by compression of a cylindrical probe (3 mm diameter) and measuring maximum compression force (i.e., when the cylindrical probe has penetrated the skin) in Newtons (N).

After measuring the firmness, every fruit was cut into two equal pieces and fruit pericarp thickness was measured by a digital caliper (CD-20APX; Mitutoyo Corp., Kanagawa, Japan). Then, every fruit was cut into small pieces and made into juice using mortar and pestle for measuring sugar content and acidity. Digital pocket refractometer ATAGO PAL-1 (Atago Co. Ltd., Tokyo, Japan) was used for measuring sugar content, and PAL-BX/ACID1 (Atago Co. Ltd., Tokyo, Japan) was used for measuring acidity in paprika.

*2.8. Statistical Analysis*

The experiment was conducted in a completely randomized design with ten single plant replicates per treatment. Effects of treatments were analyzed using SAS program (statistical analysis system, version 9.3, SAS Institute, Cary, NC, USA). Significant differences among the means were examined using analysis of variance (ANOVA) followed by Duncan's multiple range test (DMRT) at $p \leq 0.05$. OriginLab 10.0 software 176 (OriginLab, Northampton, MA, USA) was used for principal component analysis (PCA).

## 3. Results and Discussion

It was found that the temperature of different heights (floor, plant and roof) in the greenhouse was slightly higher in the S2 treatment compared to the S1 treatment (Figure 3A,B). The leaf temperature and sap flow of plants were also measured. Figure 3C,D illustrated that the leaf temperature at a 1 m height of the plant was similar in both S1 and S2-treated plants. However, at a 2 m height, it was higher in S2-treated plants compared to S1 from 11 a.m. to 1 p.m.

Sap flow is the movement of fluid in the roots, stems and branches of plants. It can be measured to identify the plants' hydrological activity and drought stress conditions. Generally, the sap flow rate is low at night time compared to daytime because, during the night, the water utilization activity of plants is low [33]. In this study, it was also observed that, in the early morning and night, the sap flow of paprika plants was low and similar in both S1 and S2-treated plants, but it was higher in S1-treated plants than those grown in the S2 treatment from 11 a.m. to 5 p.m. (Figure 3C,D). The high light intensity from 11 a.m. to 3 p.m. in the S1 treatment (Figure 1A) may be the reason for the high sap flow rate in S1-treated paprika plants, because the light intensity enhances the transpiration and other photosynthetic activity of plants [34,35] and leads to increasing the sap flow rate in plants. Researchers reported that the sap flow of plants is correlated with transpiration [36].

Furthermore, the graphs showed that the trend of the difference between the leaf and air temperature in the greenhouse was also similar in the S1 and S2 treatment (Figure 3E,F). From 9 a.m. to 12 p.m., the leaf temperature at the top portion (2 m height) of the plant was higher than the air temperature, and it was highest (4 °C) in the S2 treatment. From 12 p.m. to 10 p.m., the leaf temperature was lower than the air temperature, and this difference was highest around 6 °C at 4 pm in both S1 and S2 treatments. The weight of substrate media fluctuated and was similar in both treatments, and it was higher in both S1 and S2 treatments between 9 a.m. and 3 p.m. It fluctuated because of the supplied nutrient solution in the substrate media and the fact that transpiration occurred in the plant.

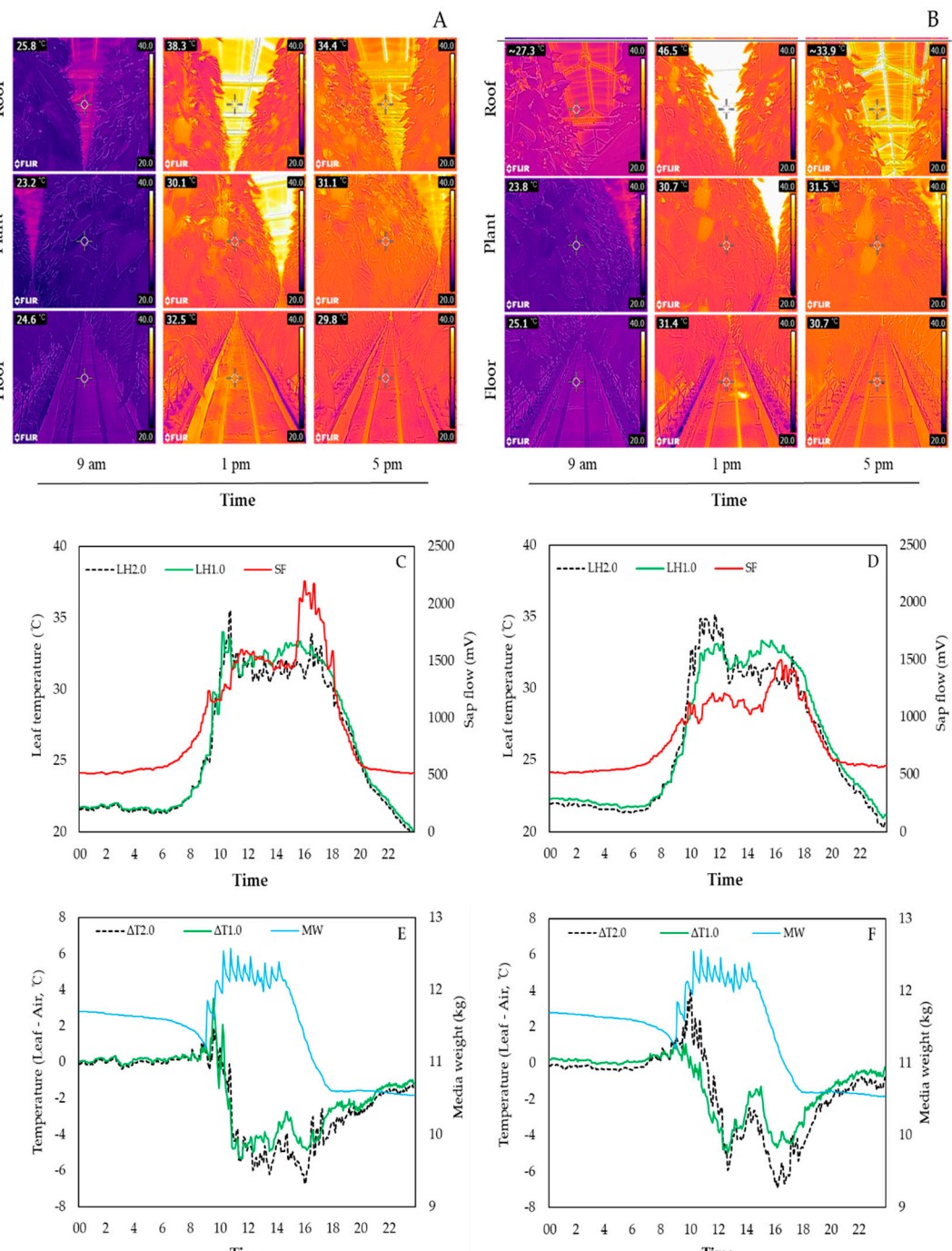

**Figure 3.** Temperature at different positions of greenhouse in S1 (**A**) and S2 (**B**) treatments. Leaf temperature and sap flow of S1 (**C**) and S2 (**D**)-treated plants. LH2, 2 m height of plant; LH1, 1 m height of plant; SF, sap flow. The temperature difference between leaf and air in S1 (**E**) and S2 (**F**) treatments. ΔT2.0, 2 m height of plant; ΔT1.0, 1 m height of plant; MW, growing media weight.

The Fv/Fm (maximum quantum yield of PSII) ratios were measured in order to know the dark-adapted state (DAS) efficiency and photochemical activities in PSII. In this experiment, a significant variation in the S1 and S2-treated plant's Fv/Fm ratio was only observed at 11 a.m. in both top and bottom leaves of plants (Figure 4A). Bjorkman and Demmig [37] state that the Fv/Fm values between 0.78–0.83 indicate healthy and non-stressed plants. With an increasing day period, the values of Fv/Fm in both treatments were good in the top leaves of paprika plants, and there was no significant variation between

the treatments. This may be the reason for why plants were adapted to light intensity in both treatments with increasing daytime in the greenhouse. Furthermore, the $PI_{ABS}$ (performance index on absorption basis) was always higher in the top leaves of plant than the bottom leaves. At 2 p.m., the $PI_{ABS}$ of upper leaves was higher in S1-treated plants than those grown in the S2 treatment (Figure 4B).

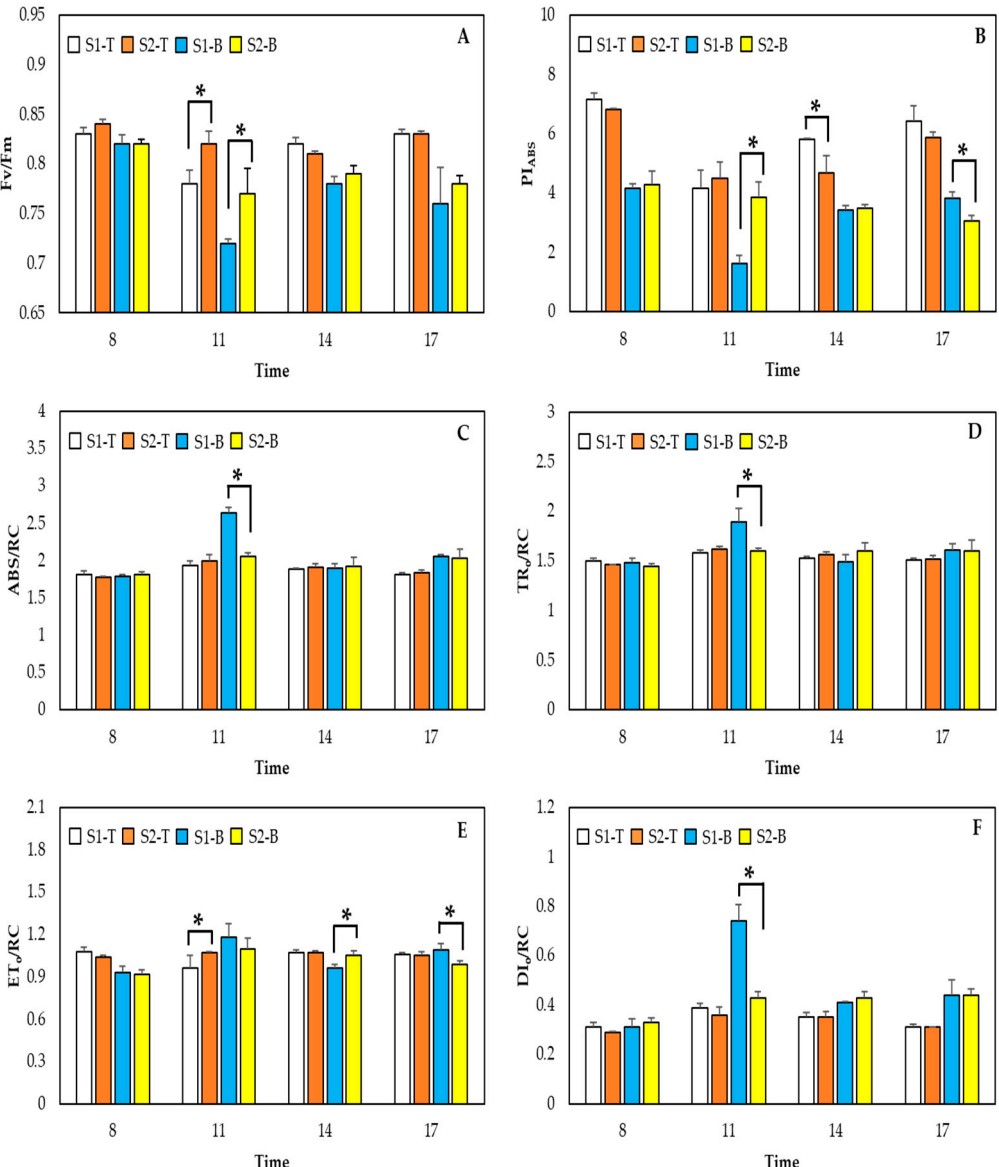

**Figure 4.** Effect of shade screen on Fv/Fm (**A**), $PI_{ABS}$ (**B**), ABS/RC (**C**), $TR_o$/RC (**D**), $ET_o$/RC (**E**) and $DI_o$/RC (**F**) of paprika plant in greenhouse. S1 and S2 indicate plants were treated by single shade screen and double shade screens, respectively. T and B indicated top portions at 2 m height and bottom portions at 1 m height of the plant, respectively. Lines above bar represent the standard deviation of the mean (*n* = 10). *, above the mean bars indicate significant difference between the treatments by Duncan's multiple range test (DMRT) at $p \leq 0.05$.

During the time, ABS/RC (absorption per active reaction center) and $TR_o$/RC (trapped energy flux per active reaction center) showed a similar trend in both parts of the plants in both treatments, except at 11 am (Figure 4C,D). At 11 a.m., the ABS/RC and $TR_o$/RC of lower leaves of plants were significantly higher in S1-treated plants compared to the S2 treatment. During the whole time period, the value of ABS/RC and $TR_o$/RC was always more than 1.5.

Moreover, the $ET_o/RC$ (electron transport flux per active reaction center) value was almost constant in the top leaves of S2-treated plants (Figure 4E). However, it showed significant variation between the bottom leaves of S1 and S2-treated plants at 2 p.m. and 5 p.m. The $DI_o/RC$ (rate of energy dissipated by PSII per reaction center) showed fluctuation in S1 and S2-treated plants (Figure 4F). A significant difference was only found at 11 a.m. in lower leaves of the paprika plant, and it was 1.5 times higher in S1-treated plants than in S2. Researchers noted that the photosynthetic efficiency of a plant is reflected by the activities of its chlorophyll fluorescence [38,39]. In addition, the photosynthetic production of leaves depends on received light irradiance, and is increased by the increasing light exposure of leaves [40].

NDVI illustrates relative reflectance as a measure of relative plant health in landscapes and terrestrial habitats by measuring the difference between near-infrared (which the plant strongly reflects) and red light (which the plant absorbs) [41]. In this study, from Figure 5, we found that a green color was higher in both S1 and S2-treated plants, which indicates in both treatments that the plants were healthy. Many researchers reported that the high green color index of NDVI indicates healthy tissue of plants [24,42,43]. Furthermore, an NDVI value greater than 0.5 represents a good physiological condition of plants. In S1 and S2 treatments, NDVI values were more than 0.70, and were highest in the S1 treatment (Figure 5E), which indicates that the light protection system in S1 is more feasible for the physiology of paprika plants.

The plant height was significantly taller (14%) in S2-treated plants than those grown in the S1 treatment (Figure 6A). Compared to the S1 treatment, an around 10% low light intensity (Figure 1A) was received by the plants, which is the reason for the tallest plants in the S2 treatment. Ha et al. [44] showed that two cultivars of paprika were grown in a greenhouse and that their plant height was significantly increased (10%) by the low light intensity. Rylski and Spigelman [45] showed that shade conditions significantly increased the plant height of paprika compared to the control. Diaz-Perez [46] reported that the plant height of greenhouse-grown paprika was significantly increased under 20% to 80% low light conditions. Furthermore, Galvez et al. [47] and Kesumawati et al. [48] reported that the shade-treated chili plant (*Capsicum annum*) was significantly taller compared to those grown in non-shade conditions. Jeeatid et al. [49] reported that the plant height of four cultivars of *Capsicum chinense* grown in a plastic net house was significantly higher in 70% low light conditions compared to the control. Tinyane et al. [50] reported that a low light intensity increased the plant height of tomato. This can be partly explained by the fact that the central hours of the reduced light intensity inside of the greenhouse led to changes in micro-climate conditions (temperature, humidity, etc.) and resulted in enhancing the plant growth elongation. This hypothesis is also supported by Galvez et al. [47] and Song et al. [51]. Another family plant, asparagus, also showed that, at the initial stage of growth, a 30% low light intensity enhanced the plant elongation in a greenhouse [52]. Usually, plants enhance their vertical growth under a low light intensity to reach more light. Researchers reported that shaded plants regulate their assimilated carbon for vertical growth in order to capture the furthest light energy [53,54].

On the other hand, the number of leaves was significantly higher (29%) in S1-treated plants compared to S2-treated plants (Figure 6B). Compared to S1, in the S2 treatment, light penetration in the lower part of the paprika plant through the canopy is too low because of the upper part of the leaf surface of the plant. Diaz-Perez [46] also reported that the number of leaves of the paprika plant decreased under low light. Many researchers reported that plants undergo morphological changes, such as a longer internode, declining number of leaves and thinner leaves for adaptation to low light, and maximize the use of their received light [55,56].

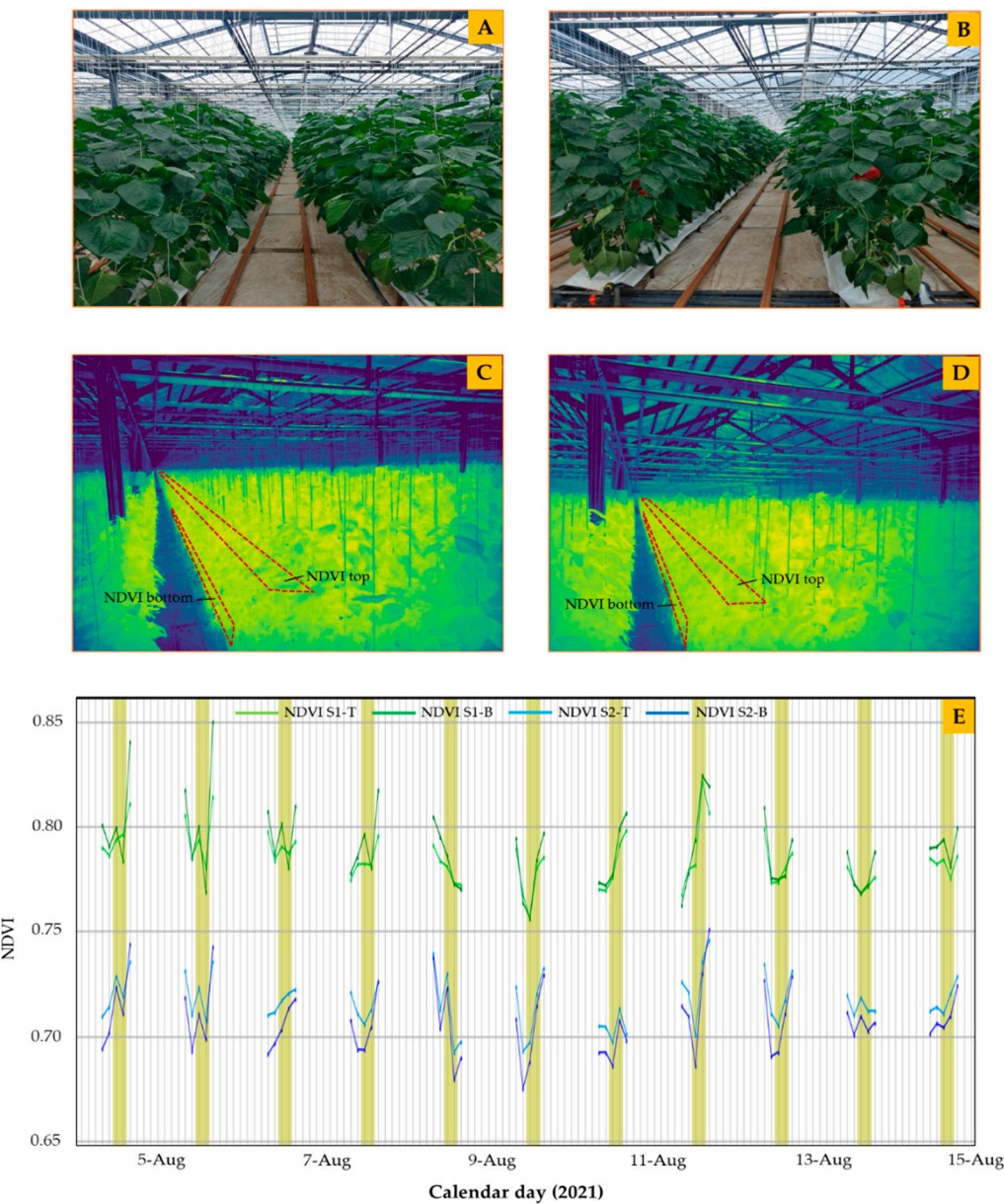

**Figure 5.** Effect of shade screens on plant growth of S1 (**A**) and S2 (**B**) treatments. Image of normalized difference vegetation index (NDVI) in S1 (**C**) and S2 (**D**)-treated plants. Graph of NDVI in S1 and S2 treatments (**E**). T and B indicate top portions at 2 m height and bottom portions at 1 m height of the plant, respectively.

In addition, leaf fresh weight and leaf dry weight were significantly higher (around 30%) in S1-treated plants than in S2-treated plants (Figure 6C,D). Increased numbers of leaves are one of the reasons for the greater fresh and dry weight of paprika plants in S1 treatment than those grown in S2. Zhu et al. [57] also reported that the leaf weight of *Capsicum annum* was significantly reduced by the high shade treatment when plants were grown at a 70–85% field moisture capacity. Other researchers reported that highly shaded plants usually transferred photosynthetic products to leaves for growth activity; however, this partially compensates for the decreased growth rate because of the reduced light energy [53,54].

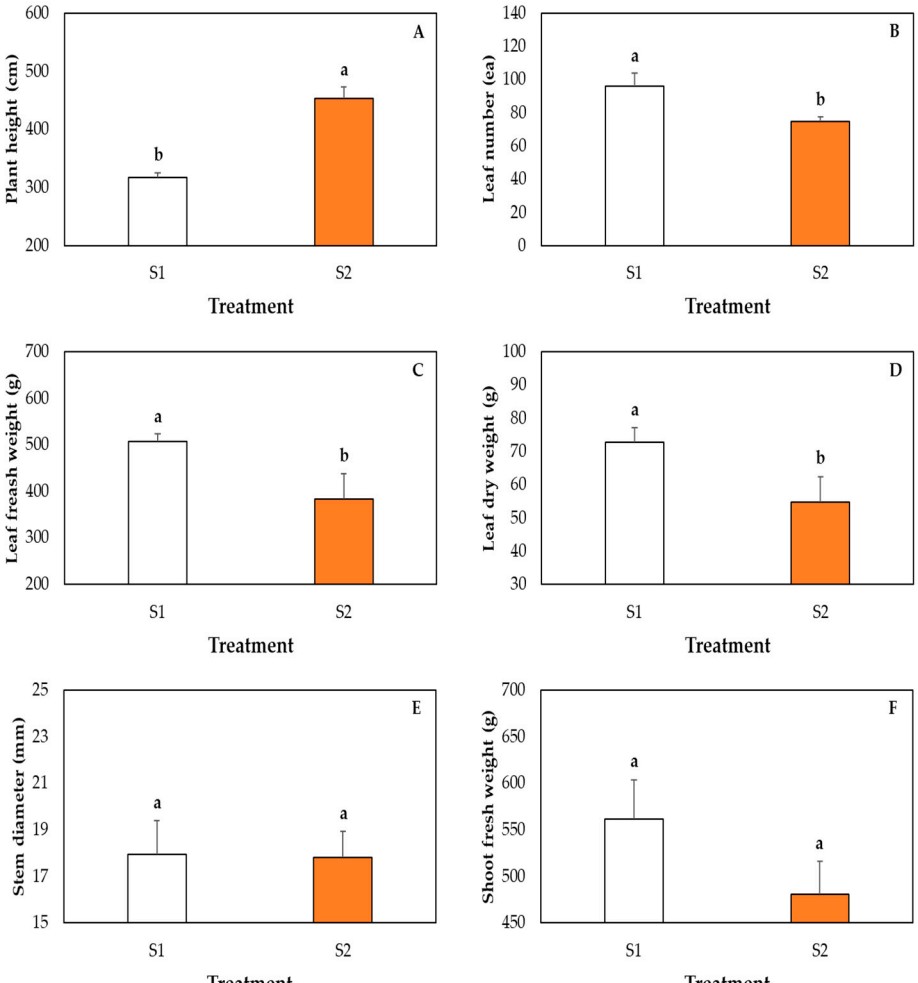

**Figure 6.** Effect of shade screen on plant height (**A**), leaf number (**B**), leaf fresh weight (**C**), leaf dry weight (**D**), stem diameter (**E**) and shoot fresh weight (**F**) of paprika in greenhouse. S1 and S2 indicate that plants were treated by single shade screen and double shade screens, respectively. Lines above bar represent the standard deviation of the mean (*n* = 10). Means above each bar followed by the same letters are not significantly different by Duncan's multiple range test (DMRT) at $p \leq 0.05$.

The stem diameter and shoot fresh weight were not significantly different between S1 and S2-treated plants (Figure 6E,F). It is possible that the duration of the growing period and the difference in light intensity between the treatments were not enough to significantly impact the stem diameter and total shoot fresh weight of the plant. Other researchers also found similar results in their experiments. Ha et al. [44] and Diaz-Perez [46] reported that the stem diameter of paprika plants grown in a greenhouse was not significantly different between the shade treatment and the control. Zhu et al. [57] reported that the shoot weight of *Capsicum annum* was not significantly different between plants treated with 0%, 30% and 50% low light intensity grown at a 75–85% field moisture capacity.

The number of fruits was significantly higher (39%) in S1-treated plants (Figure 7A). Rylski and Spigelman [45] showed that the number of fruits of paprika was significantly (33%) increased in control conditions compared to those grown in 40% shade. Castronuovo et al. [58] reported that the number of fruits per plant of paprika was significantly higher in the control condition than those plants treated with 30% and 50% low light intensity. Zhu et al. [57] reported that the yield of *Capsicum annum* was significantly reduced under a low light intensity.

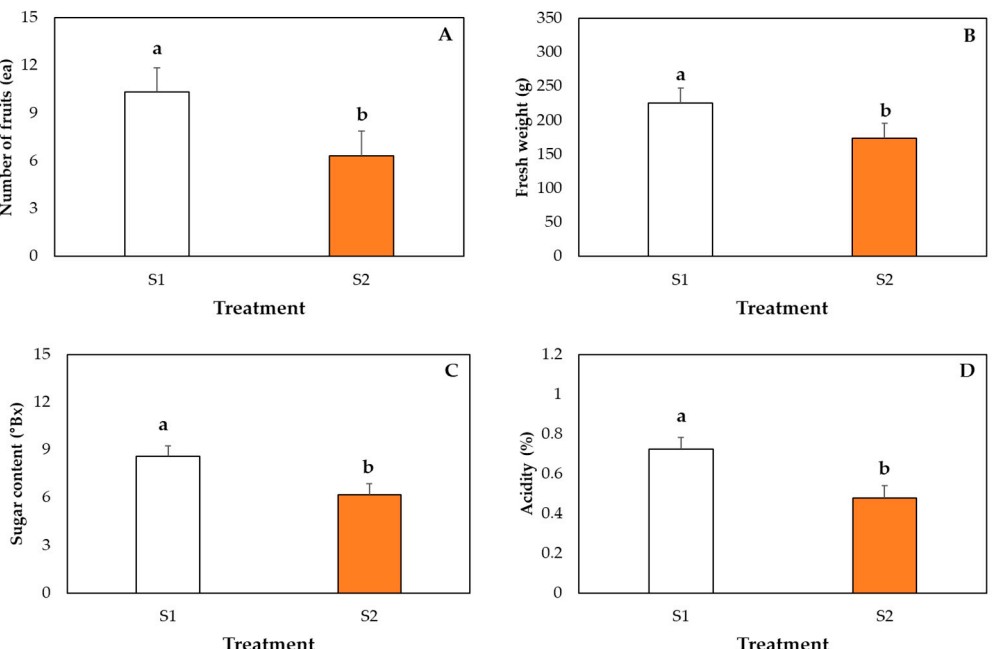

**Figure 7.** Effect of shade screen on number of fruits (**A**), fruit weight (**B**), sugar content (**C**) and acidity (**D**) of paprika in a greenhouse. S1 and S2 indicate that plants were treated by single shade screen and double shade screens, respectively. Lines above bar represent the standard deviation of the mean (*n* = 10). Means above each bar followed by the same letters are not significantly different by Duncan's multiple range test (DMRT) at *p* ≤ 0.05.

The fruit fresh weight was significantly higher (30%) in S1-treated plants compared to the S2 treatment (Figure 7B). The fruit length, fruit width and fruit diameter of mature fruits were not significantly different between S1 and S2-treated plants (Table 2). However, the pericarp thickness of S1-treated paprika was significantly higher (31%) than those grown in the S2 treatment. In addition, the volume of S1-treated fruit was significantly higher (26%) than S2-treated fruit (Table 2). The fruit length, fruit width and fruit diameter were not significantly different but were slightly higher in S1-treated fruits and resulted in a cumulative impact on the fruit volume. For this reason, the fruit volume was significantly higher in S1-treated plants compared to S2-treated plants, and this finding is an important finding of this research (Table 2). In addition, an increased volume and pericarp thickness are the reasons for the higher fruit fresh weight in S1-treated fruits compared to those grown in the S2 treatment.

**Table 2.** Effect of shade screen on fruit growth characteristics of paprika in greenhouse. S1 and S2 indicate that plants were treated by single shade screen and double shade screens, respectively.

| Treatment | Length (cm) | Width (cm) | Diameter (cm) | Pericarp Thickness (mm) | Firmness (N/φ5 mm) | Hunter Value | | | Volume (cm³) |
|---|---|---|---|---|---|---|---|---|---|
| | | | | | | L | a | b | |
| S1 | 10.2 ± 0.5 [z] a [y] | 8.6 ± 0.6 a | 27.1 ± 1.9 a | 7.6 ± 1.7 a | 17.65 ± 0.81 a | 33.8 ± 5.6 a | 30.7 ± 4 b | 17.18 ± 9.8 b | 763.5 ± 122.3 a |
| S2 | 9.5 ± 0.6 a | 8.0 ± 0.2 a | 25.1 ± 1.5 a | 5.8 ± 0.8 b | 13.28 ± 2.06 b | 30.7 ± 3.2 a | 42.5 ± 5.2 a | 28.12 ± 3.1 a | 603.6 ± 37.3 b |

[z] Each value is the mean (*n* = 10). ± indicates the standard deviation of the mean. [y] Means within columns sharing the same letter are not significantly different based on Duncan's multiple range test at *p* ≤ 0.05.

The fruit firmness of S1-treated paprika was significantly higher (33%) than S2-treated plants (Table 2). The increased pericarp thickness may be the reason for the increased firmness in S1-treated fruit compared to S2-treated fruit. Researchers noted that the firmness of fruit depends on different enzyme activities [59,60]. Khan et al. [61] reported that the ethylene content increment of fruit under different conditions is responsible for tissue softening and for reducing the firmness of the fruit. On the other hand, in maximum studies, researchers found changes in the firmness of different fruits during storage

conditions [3,62,63]. Therefore, the observed significant variation in paprika firmness between the S1 and S2 treatment in our experiment might have been an instant measurement after harvesting. In addition, the shade level and temperature difference are the reasons for enzymatic and non-enzymatic activities in the impact on the firmness of paprika at the maturity stage. The sugar content was significantly higher (28%) in S1-treated paprika compared to those harvested from S2-treated plants (Figure 7C). The acidity content was also significantly higher in S1-treated paprika than S2 (Figure 7D).

Color is an important parameter of paprika for consumer preference. In this study, we found that Hunter value *L* was not significantly different between the treatments (Table 2). On the other hand, Hunter values *a* and *b* were significantly higher in S2-treated fruits. Earlier studies showed that the redness and yellowish of paprika is related to the presence of various bioactive compounds, such as carotenoids, polyphenols and flavonoids [1,6]. In addition, Almela et al. [64] reported that a growing temperature influences the color accumulation of paprika, and the intensities of yellowish and reddish hues change. During the daytime, the temperature of the S2 treatment was slightly higher than the S1 treatment. This temperature difference might be the reason for the lower Hunter values *a* and *b* of S1-treated paprika compared to S2-treated paprika.

The principal component analysis (PCA) was also implemented to uncover the correlation of the different growth parameters and fruit characteristics of paprika with the different shade screen treatments (Figure 8). This PCA biplot represents clear segregation into two clusters among the parameters. The graph indicates that the number of leaves and leaf fresh weight are positively correlated. The fruit fresh weight and fruit volume also have a positive correlation, and their response is closer to the S1 treatment. Furthermore, the result showed that the sugar content and acidity were positively correlated. Other research also showed that the sugar content and acidity in paprika have a positive correlation [6]. This hypothesis is supported by the present findings.

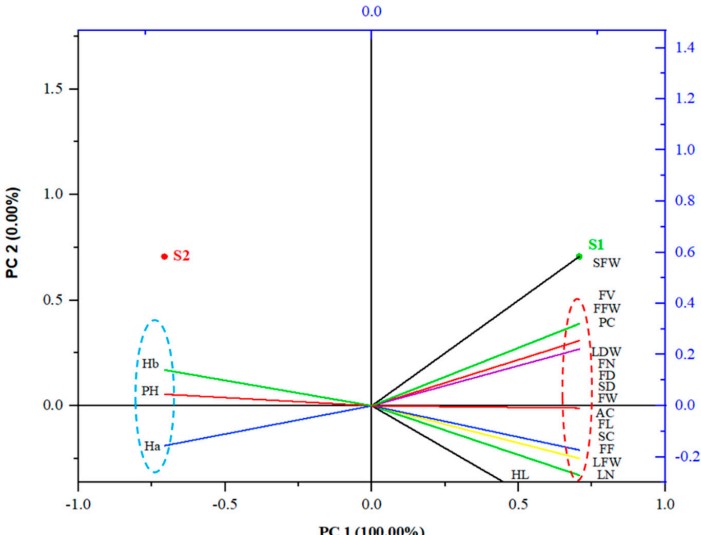

**Figure 8.** Principal component analysis (PCA) illustrates the variable treatment relationships between the treatments of plant growth and fruit characteristics of paprika. S1 and S2 indicate that plants were treated by single shade screen and double shade screens, respectively. The lines starting from the central point of the biplots display the negative or positive associations of the different variables, and their proximity specifies the degree of correlation with specific treatment. PH, plant height; LN, number of leaves; LFW, leaf fresh weight; LDW, leaf dry weight; SD, stem diameter; SFW, shoot fresh weight; FN, number of fruits; FFW, fruit fresh weight; FD, fruit diameter; FL, fruit length; FW, fruit width; PC, pericarp thickness; FF, fruit firmness; FV, fruit volume; SC, sugar content; AC, acidity; H-L, Hunter L; H-a, Hunter a; H-b, Hunter b.

## 4. Conclusions

Because of different light intensities, the growing environment showed slight changes between S1 and S2 treatments. Although the plant height was taller and the Hunter values '*a*' and '*b*' were higher in S2-treated plants, this treatment is not preferable for the plant growth and fruit production of paprika compared to S1. The physiological activity sap flow and NDVI were better in S1-treated plants. The fruit fresh weight, number of fruits, fruit volume, sugar content and acidity were significantly higher in S1-treated plants, where more sunlight was allowed inside of the greenhouse, than in S2. In future research, we will investigate the impact of single-screen and double-screen shade on bioactive compounds of paprika that will be grown in the greenhouse.

**Author Contributions:** K.Y.C. designed the experiment and supervised the study. K.H.K. and M.R.A.S. carried out all of the experimental works, analyzed the data and drafted the final manuscript. J.H.A., H.J.L., D.J.K., I.-C.H. and J.H.B. collected and analyzed the data. All authors have read and agreed to the published version of the manuscript.

**Funding:** This research was supported by the MSIT (Ministry of Science and ICT), Korea, under the ITRC (Information Technology Research Center) support program (IITP-2021-2018-0-01433) supervised by the IITP (Institute for Information & communications Technology Promotion).

**Institutional Review Board Statement:** Not applicable.

**Informed Consent Statement:** Not applicable.

**Data Availability Statement:** Not applicable.

**Acknowledgments:** This work was supported by the Korea Institute of Planning and Evaluation for Technology in Food, Agriculture and Forestry (IPET) through the Agri-Food Export Business Model Development Program (320101033HD030) and Korea Smart Farm R&D Foundation of Korea (KoSFarm) (421009-04) funded by Ministry of Agriculture, Food and Rural Affairs (MAFRA).

**Conflicts of Interest:** The authors declare no conflict of interest.

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
