# Peer review of "Effect of Shade Screen on Sap Flow, Chlorophyll Fluorescence, NDVI, Plant Growth and Fruit Characteristics of Cultivated Paprika in Greenhouse"

_agriculture, doi:10.3390/agriculture12091405_

Round 1

Reviewer 1 Report

The manuscript is interesting and contains a lot of valuable information. The aim of the research was to investigate the effect of shade screens on physiological activity, growth parameters and fruit characteristic of paprika plant.

The title accurately reflect the content of the article, but I suggest to remove “summer” from it. Effect of shade screen on sap flow, chlorophyll fluorescence, NDVI, plant growth and fruit characteristics of paprika cultivated in greenhouse.

Besides, I have a few minor comments.

Line 62-62 – What did the authors mean by "photosynthetic metabolism”?

Line 380 – The table 2. should be stretched across the width of the page. It will be more readable.

Line 382 – What does "z" and "y" mean in table footnotes?

Line 442 - Food Chem.

Line 458 – no source

Line 479-480, 557 – journal title in italics

Line 559 – Food Chem.

Line 561 – J. Agric. Food Chem.

Author Response

Thank you for your review. We are edited our manuscript including your review points.

Reviewer 2 Report

The authors studied the effect of single or double layer light screens on plant physiological parameters, growth, and fruit morphology and quality of paprika (pepper).

The introduction is informative and sufficiently explains the scope of the research. The materials and methods are clear. The overall findings (under S2 there were taller plants, lower photosynthetic efficiency displayed by fluorescence parameters, lower fruit quality) indicate that S2 provided too much light screen, while S1 is more appropriate to increase the production and quality of paprika.

I would like the authors to explain why they did not include a control without light screen, as well as how they selected these two light screens. It is possible that a light screen obstructing less light than S1 would perform even better compared to S1.

Specific comments are following.

·         L36-47. Please, provide some economic information about paprika production in Korea and the rest of the world.

·         L50-52. Do you have light intensity values to refer to? You may refer to the latitude of Korea so that the reader knows how much light reaches the surface in the summer months.

·         L70-71. NDVI abbreviation must be explained in the first mention, in L67 instead of here.

·         L82-83. References of these few studies?

·         L98-99. The screens were installed only between 11 am and 3 pm? They were removed before and after those times?

·         L111. In Figure 1C, you may remove the decimal on the y axis.

·         L210-212. I would expect that S2 would lead to lower temperatures. Do you have a theory about this?

·         L244-245. The Bjorkman and Demmig (1987) paper states that Fv/Fm values between 0.78-0.83 indicate healthy plants, not 0.75. Please revise.

Author Response

Thank you for your review. We edited our manuscript including your review points.

Round 2

Reviewer 2 Report

The authors sufficiently addressed my comments and suggestions. Therefore, I suggest publication of the manuscript in Agriculture. Thank you for giving me the opportunity to assess the manuscript.